# The effect of language discordance on the experience of palliative care: A scoping review

**Shawn P. Dookie**[1]*, **Lynn Martin**[1,2]

**1** Department of Health Sciences, Lakehead University, Thunder Bay, Ontario, Canada, **2** Centre for Education and Research on Aging and Health, Lakehead University, Thunder Bay, Ontario, Canada

* spdookie@lakeheadu.ca

## Abstract

### Background

Internationally, research demonstrates consistent disparities in access to palliative care services for those in underserved communities with life-limiting conditions.

### Objectives

This outcomes-oriented scoping review examines the impact of language discordance on palliative care (PC) experiences. It seeks to answer the question: what are the PC experiences of individuals who do not speak the same language as their care provider? The review explored the range and depth of existing research and synthesizing trends across studies.

### Design

Online databases were used to find articles published in English or French, from January 1, 2010, to February 5, 2024. Inclusion criteria included studies that explored the relationship between palliative, end-of-life or hospice care, as well as advance care planning in the context of language discordance between individual and health care provider. This scoping review was originally designed to explore Canadian official language minority communities, but was broadened to an international search for a more robust dataset.

### Results

A total of 39 articles were included in the review, 23 qualitative studies, nine quantitative studies and seven mixed-methods studies. The following elements were extracted from eligible articles: country, study design, target population and definitions, participant characteristics, definitions of PC, outcomes studied, findings related to the study aims, as well as author-defined study limitations and next steps. Areas for further research were identified, as were areas for policy and practice change. Studies used various perspectives of PC, often synonymizing it with end-of-life and hospice care, as well as advanced care planning. There was no contextual definition of language barriers in the studies and

**Data availability statement:** All relevant data are within the paper.

**Funding:** The production of this scoping review has been made possible through a financial contribution from Health Canada. The views expressed herein are the views of the authors, and do not necessarily represent the views of Health Canada. The funders had no role in the study design, data collection, analysis or preparation of the final manuscript.

**Competing interests:** The authors have declared that no competing interests exist.

no studies that explored the PC in any Canadian official language minority communities. The importance of in-language resources, accessibility of skilled interpreters, education in cross-cultural care were all common themes in the literature.

## Conclusions

From a variety of perspectives, studies generally found that language discordance has a negative impact on the quality of accessibility of palliative, end-of-life and hospice care, as well as advanced care planning. Given that Canada's population is becoming increasingly linguistically diverse, there is a need to better understand the impact of receiving PC from individuals who speak another language on both the quality of PC and quality of life at the end of life.

## Background

Palliative care (PC) is an interprofessional approach to care for individuals with life-limiting health conditions that focuses on autonomy, comfort, holism, and dignity [1]. Providing equitable health care to an ethnoculturally diverse community can be difficult [2]. Studies around the globe have identified inequitable access to PC, where socio-cultural and racial differences play a significant role in these disparities [3]. In Canada, there is a strong push to facilitate equitable access to quality PC. Current research and policy development efforts focus on identifying gaps in professional training, increasing public awareness, and working to meet the needs of the ever-changing, socio-cultural diversity of Canadian communities [1]. As language barriers are known to have a negative impact on access to and quality of health care, as well as outcomes, increased resources have been dedicated to finding ways to mitigate such disparities [4].

As per the *Official Languages Act* of 1985, Canadians can access health services in either of the two official languages, English and French [5]. There are geographic regions throughout the country where one of the official languages is more often used than the other [6], for example a French-speaking individual living in a predominantly English-speaking community (or vice versa) is considered part of an official language minority community (OLMCs) [7]. In Ontario (Canada), medical translators, terminologists, and interpreters are regulated [8]. Certified Medical Interpreters (CMIs) receive training in medical terminology and pass a national standardized exam [9].

While members of OLMCs have a legal right to services in their official language, allophones – i.e., those with a mother tongue other than English or French, are not afforded the same right [10]. This is especially problematic in Ontario, where about 28% of the population (3.7 million individuals) are allophones. Ontarians have identified 200 different languages as their mother-tongue, the most common being Chinese languages (primarily Mandarin or Cantonese), though Italian, Punjabi, Spanish, Arabic, Tagalog, Urdu, and Portuguese are also common [11]. Individuals with language discordance, including allophone newcomers to Canada and members of OLMCs, are often underserved, and at higher risk of inadequate end-of-life planning and death in-hospital [12]. Immigration to Ontario is at an all-time high, and while the COVID-19 pandemic impacted these numbers, early estimates project that the province will welcome about 220,000 immigrants in the 2025–26 census year [13]. As such, it is safe to assume that Ontario communities will become even more linguistically diverse.

This exploratory, outcomes-oriented scoping review [14] seeks to answer our research question: what are the PC experiences of individuals who do not speak the same language as

their care provider? The narrative describes the impact of language discordance on PC from the perspective of linguistic minorities, explores the range and scope of existing research, and synthesizes trends across studies.

## Methods

Synthesis of the literature was conducted using a scoping approach, as it aligned well with our research goal of exploring and mapping out the current literature. With the goal of identifying and synthesizing the broad, heterogeneous range of study designs and findings, our scoping review will be used to inform future research priorities in language discordance in palliative care [14,15]. To provide a more structured methodological approach, the Preferred Reporting Items for Systematic reviews and Meta-Analyses extension for Scoping Reviews (PRISMA-ScR) S1 Checklist was used to guide reporting of the scoping review procedures and findings [15].

### Data sources

The literature search was conducted using PubMed, Web of Science, and Canadian Business and Current Affairs (CBCA) databases. Results were limited to full-text peer-reviewed articles published in English or French, from January 1, 2010, to February 5, 2024. Review articles were excluded. In consultation with a research librarian, two searches were conducted. A narrowly focused search was done on January 20, 2024, using the following terms: "Linguistic Minority" OR "Linguistically diverse" OR CaLD OR "Cultural and Linguistic Diversity" AND "Palliative Care" OR EOL OR "End-of-Life Care", followed by a broader search on February 5, 2024: "Racial Minorit*" OR Racial*OR "Cultural Diversit*" OR "Ethnic Diversity" AND "Palliative Care" OR EOL OR "End-of-Life Care". The goal was to identify similar studies of underserved populations, specific to racialized communities, and identify studies that examined various ethno-cultural intersections including language discordance.

### Eligibility criteria

The first author led the search, selection process, and data extraction with the support of a Librarian and the second author. Using the mutually developed search terms, the first author performed initial searches and title reviews, documenting reasons for exclusion. The second author reviewed all decisions; any disagreements were discussed and decisions (to include or exclude) were approved by both. The same process was followed through the abstract reviews. We did not experience any disagreements in decision-making and tended to air on the side of inclusion if we questioned the relevance of an article.

Studies that explored palliative care, end-of-life care, advanced care planning, and hospice care, or any variations of these terms, were included. As previously discussed, there is an important distinction between these concepts, but as evidenced through my initial test literature searches, there is paucity of research specifically exploring palliative approaches to care in linguistic minorities, so this term was broadened. Articles from countries with similar socio-economic standing (e.g., OECD countries) were accepted. Articles that explored linguistic minorities, linguistically diverse communities and language barriers were accepted in the first search, this was expanded to accept studies that explored ethno-cultural diversity, if the study analyzed the impact of language on the experience of palliative or end-of-life care.

Studies that lacked information on the target population, studies of palliative care where language concordance was not explored, and studies that focused on curative therapies (as opposed to palliative therapies) were not considered. Literature reviews and scoping reviews

were not included in the final selections, but their reference lists were reviewed for any articles missed in the search.

### Data charting and data items

Data charting and extraction was completed using a Microsoft Excel Spreadsheet. This scoping review is part of a much larger project exploring palliative the care experiences of underserved populations, and as such, we initially worked with a common set of data items for extraction: country; target population; study design, setting, participants, and characteristics; terminology and definition of care received; outcomes studied; and author-defined study limitations and next steps. The first author completed the extraction for the first ten articles (in alphabetical order by first author), and these were fully reviewed by the second author and no changes were made to the extraction. The second author completed extraction of the remaining articles. The first author identified a set of initial codes for summarizing key elements in the extracted information, which were reviewed by the second author. Inconsistences and discrepancies identified were discussed and resolved to the satisfaction of both authors. The data extraction summary included below represents the final collaborative findings of both authors.

## Results

The searches yielded 594 articles for title review (after elimination of duplicates), of which 209 were kept for screening. Sixty-six articles were fully read, and 39 were included in the data extraction (Fig 1).

### Characteristics of sources of evidence

This section characterizes the sources of evidence based on data items extracted from each of the eligible 39 articles (Table 1).

**Country.** Of the 39 studies, fifteen came from Australia [16–30], twelve from the United States [31–42], seven from Europe and the United Kingdom [43–49], and five from Canada [50–54].

**Target population.** In the literature review, 13 articles did not specify the language group studied [22,31,34,35,37,41,42,46–49,51,54]. For example, Cicolello and Anandarajah studied "members of ethnic and racial minority populations" [31, p870]. The article by Yarnell and colleagues focused on "recently immigrated versus long-standing [Canadian] resident decedents" [54, p1480]. While they found an association between language proficiency and end-of-life care, they did not define specific languages, ethnic groups, or countries of origin. Javanparast and colleagues explored Australian PC policy in the context of priority underserved populations, including those with language discordance, but do not specify which language(s) [22].

The Australian Bureau of Statistics (ABS) has clearly defined Standards for Statistics on Cultural and Language Diversity (CALD) to facilitate data collection and analysis. There are four main concepts used to define CALD: (1) country of birth, (2) language spoken at home, (3) English proficiency and (4) Indigenous status [55]. Critiques are that the term CALD has been overused as a blanket statement and has been misused at times – labeling groups based on race or ethnicity, without considering Aboriginal status, country of birth, or language proficiency [56]. Nine of the 15 articles reviewed from Australia used only the term CALD without further specifying a target population based on ABS criteria [17,19–21,23–25,27,28]. Five Australian articles discussed how their target population met CALD standards per the ABS [16,18,26,29,30]. For example, Leonard and colleagues narrowed their focus on

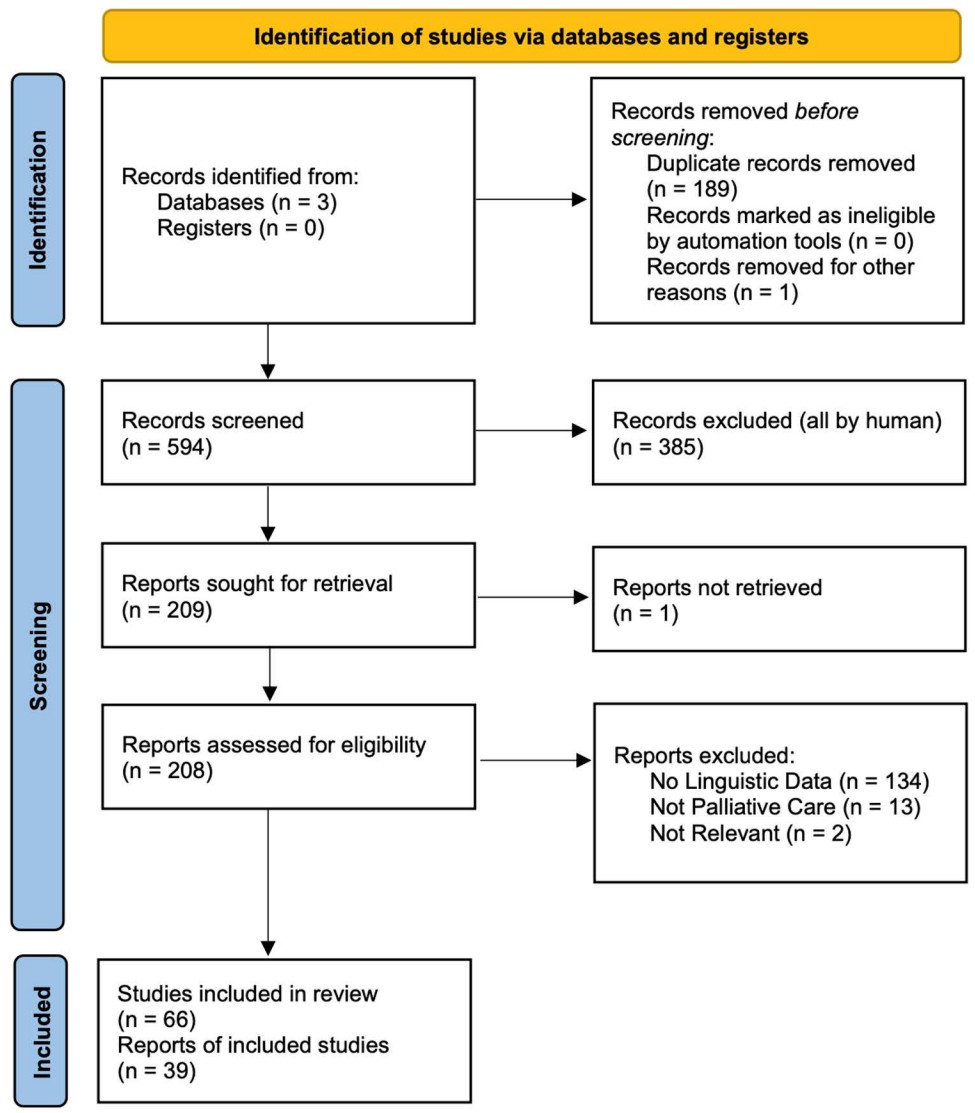

**Fig 1. PRISMA 2020 flow diagram for new systematic reviews which included searches of databases and registers.**

Australians who speak Arabic, Hindi, Mandarin, or Australian Aboriginal languages at home [26]. Bloomer and colleagues categorized their target CALD groups in relation to religious affiliation, not language [16].

Beyond the five Australian studies of specific CALD communities discussed above, 12 articles clearly defined the population of study. Here, seven studies identified the target population based on the regional demographics [39,40,43–45,50,53]. For example, de Voogd and colleagues explored the experiences of the three largest migrant groups in the Netherlands: Turkish, Moroccan, and Surinamese [44]. Eckemoff and colleagues, on the other hand, studied Russian-Americans as the first author had a connection to this community [32].

Four studies synonymized language with country or region of origin [33,36,38,52]. Glaser and colleagues focused on individuals with limited English proficiency born in Latin America but did not specify languages spoken [33]. Similarly, Khosla and colleagues did not specify a

**Table 1. Summary of data extraction.**

| # | Reference | Country | Design | Target | Setting | Sample Size | Participants | Participant Info | Term | Findings - themes |
|---|---|---|---|---|---|---|---|---|---|---|
| 16 | Bloomer et al. (2019) | AU | Mixed Methods | CALD | Inpatient rehab facility | 73 | Patients, clinicians | Age; Occupation | EOL | Cultural competence; Provider experience; Culture & care; Discordance & quality |
| 17 | Broom et al. (2013) | AU | Qualitative | CALD | Other/Not Defined | N/A | Clinicians, Professionals | Occupation | PC | Culture & care; Interpreters; Discordance & quality |
| 18 | Detering et al. (2015) | AU | Quantitative | CALD | Hospital | 112 | Patients | Age; Sex; Language | ACP | Interpreters; Culture & care |
| 19 | Green et al. (2018) | AU | Qualitative | CALD | Community PC | 28 | Clinicians | Occupation | EOL | Culture & care; Interpreters; Provider experience; Discordance & quality |
| 20 | Green et al. (2019) | AU | Mixed Methods | CALD | Community PC | 100 | Patients | Age; Sex; Birth country | PC | Discordance & quality; Cultural competence; Culture & care; Interpreters |
| 21 | Hayes et al. (2020) | AU | Qualitative | CALD | Hospital | 33 | Patients | Occupation | ACP | Discordance & quality; Culture & care; Interpreters |
| 22 | Javanparast et al. (2022) | AU | Mixed Methods | Underserved and priority populations | Other/Not Defined | 5 | Professionals | Occupation | PC | Cultural competence; Culture & care; Resources |
| 23 | Johnstone et al. (2016) | AU | Qualitative | CALD | Hospital | 22 | Clinicians | Sex; Occupation | EOL | Culture & care; Provider experience; Discordance & quality |
| 24 | Kirby et al. (2017) | AU | Qualitative | CALD | Hospital | 20 | Patients | Sex | PC | Interpreters; Culture & care; Discordance & quality |
| 25 | Kirby et al. (2018) | AU | Qualitative | CALD | Hospital | 30 | Patients, Caregivers | Sex; Language | PC | Discordance & quality; Culture & care |
| 26 | Leonard et al. (2023) | AU | Mixed Methods | CALD | Community | 306 | General community | Sex; Migrant Group | EOL | Interpreters; Resources; Culture & care; Discordance & quality |
| 27 | Martin et al. (2022) | AU | Qualitative | CALD | Hospital | 25 | Clinicians | Sex; Language; Occupation | PC | Cultural competence; Interpreters; Discordance & quality |
| 28 | Roydhouse et al. (2023) | AU | Quantitative | CALD | Hospital | 61,000 | Patients | Birth Country; Language | PC | Discordance & quality |
| 29 | Wardle & Bennett (2021) | AU | Qualitative | Punjabi Indians | Aged Care Homes | 27 | LTC residents, Staff | Age; Ethnicity; Occupation | EOL | Culture & care; Cultural competence; Discordance & quality |
| 30 | Yap et al. (2018) | AU | Qualitative | CALD | Community | 30 | General community | Age; Sex; Birth country | ACP | Resources; Culture & care |
| 31 | Cicolello & Anandarajah (2019) | US | Qualitative | Racial/ ethnic minority | Community Hospice | 22 | Clinicians, Professionals, Caregivers | Ethnicity; Race; Language; Occupation | HOS | Culture & care; Interpreters; Cultural competence; Discordance & quality |
| 32 | Eckemoff et al. (2018) | US | Qualitative | Russian Immigrants | Community | 13 | Patients, Offspring, Staff | Age; Sex; Birth country; Education; Occupation | EOL | Culture & care; Provider experience; Interpreters; Cultural competence; Discordance & quality |
| 33 | Glaser et al. (2020) | US | Qualitative | Latina immigrants | Community | 13 | Patients | Age; Birth country; Household income | Other | Discordance & quality |
| 34 | Hughes & Vernon (2020) | US | Qualitative | Racial/ ethnic minority | Other/Not Defined | 41 | Administrators | Occupation | HOS | Interpreters; Cultural competence; Culture & care |
| 35 | Hughes et al. (2021) | US | Qualitative | Racial/ ethnic minority | Other/Not Defined | 22 | Administrators | Occupation | HOS | Cultural competence; Interpreters |

*(Continued)*

**Table 1.** (Continued)

| # | Reference | Country | Design | Target | Setting | Sample Size | Participants | Participant Info | Term | Findings - themes |
|---|-----------|---------|--------|--------|---------|-------------|--------------|------------------|------|-------------------|
| 36 | Khosla et al. (2019) | US | Mixed Methods | South Asians | Community | 57 | Clinicians, Professionals | Age; Sex; Ethnicity; Occupation | Other | Resources; Culture & care |
| 37 | Lindley et al. (2017) | US | Quantitative | Language Barriers | Community Hospice | 1,251 | Organizations | N/A | HOS | Provider experience; Resources; Interpreters |
| 38 | McLean et al. (2016) | US | Quantitative | Latino | Community | 48 | Patients | Age; Birth country; Language; Education | ACP | Resources |
| 39 | Neiman (2019) | US | Qualitative | Hmong | Hospital | 34 | Clinicians | Sex; Ethnicity; Occupation | PC | Interpreters; Culture & care; Cultural competence |
| 40 | Pan et al. (2015) | US | Quantitative | Asian or Hispanic descent | Community | 604 | General community | Age; Sex; Language | HOS | Resources |
| 41 | Wright et al. (2013) | US | Quantitative | Racial/ ethnic minority | Hospital | 171 | Patients | Age; Sex; Ethnicity; Language | EOL | Discordance & quality; Culture & care |
| 42 | Zapata et al. (2024) | US | Quantitative | Racial/ ethnic and linguistic minority | Hospital | 51 | Patients | Ethnicity | PC | Resources; Provider experience; Culture & care |
| 43 | De Graaff et al. (2012) | NL | Qualitative | Migrants of Turkish or Moroccan descent | Community; Hospital | 83 | Clinicians, Professionals | Sex; Occupation; relationship to patient; Ethnicity | PC | Interpreters; Discordance & quality; Culture & care; Cultural competence |
| 44 | de Voogd et al. (2020) | NL | Mixed Methods | Migrants | Community PC | 136 | Patients | Age; Sex; Ethnicity | PC | Cultural competence; Resources; Discordance & quality |
| 45 | Dharni et al. (2011) | UK | Mixed Methods | South Asian Origin | Community | 55 | Patients | Age; Language | Other | Discordance & quality; Culture & care; Cultural competence |
| 46 | Milberg et al. (2016) | SE | Qualitative | Racial/ ethnic minority | Varied | 60 | Clinicians | Age; Sex | EOL | Provider experience; Discordance & quality; Interpreters; Culture & care; Cultural competence |
| 47 | Schrank et al. (2017) | AUST | Qualitative | Racial/ ethnic minority | Hospital | 21 | Clinicians | Age; Sex; Birth country; Occupation | PC | Cultural competence; Culture & care; Discordance & quality |
| 48 | Semlali et al. (2020) | CH | Qualitative | CALD | Other/Not Defined | 26 | Clinicians, Professionals | Occupation | PC | Culture & care; Cultural competence; Resources |
| 49 | Weber et al. (2021) | CH | Quantitative | Migrant | Hospital Outpatient Community | 204 | Clinicians, Professionals | Sex; Language; Birth country; Occupation | PC | Cultural competence; Discordance & quality; Resources; Interpreters; Culture & care |
| 50 | Hordyk et al. (2017) | CA | Qualitative | Nunavik | Community | 103 | Clinicians, Professionals | Occupation | EOL | Interpreters; Cultural competence; Culture & care; Discordance & quality; Provider experience |
| 51 | Jovanovic (2012) | CA | Qualitative | Racial/ ethnic minority | Community Hospice | 14 | Patients | Age; Sex; Ethnicity; Education; Language | HOS | Discordance & quality; Cultural competence; Provider experience |
| 52 | Nielsen et al. (2015) | CA | Qualitative | Chinese immigrant | Community | 12 | Individuals, Caregivers, Clinicians | Age; Sex | PC | Culture & care; Cultural competence; Provider experience; Discordance & quality |

*(Continued)*

**Table 1.** (Continued)

| # | Reference | Country | Design | Target | Setting | Sample Size | Participants | Participant Info | Term | Findings - themes |
|---|---|---|---|---|---|---|---|---|---|---|
| 53 | Vincent et al. (2019) | CA | Qualitative | Indigenous clients | Hospital | 133 | Clinicians | Occupation; Years of practice | PC | Culture & care; Cultural competence; Discordance & quality; Provider experience; Interpreters |
| 54 | Yarnell et al. (2017) | CA | Quantitative | Recent Immigrants | Hospital | 967,013 | Patients | Age; Sex; Urban/Rural; Socioeconomic status | EOL | Discordance & quality |

AU = Australia; AUST = Austria; CA = Canada; CH = Switzerland; NL = The Netherlands; SE = Sweden; UK = United Kingdom; US = United States.

ACP = Advance care planning; EOL = End-of-life care; HOS = Hospice care; PC = Palliative care.

language, they identify their target population as "…people with origins from India, Pakistan, Bangladesh, Nepal, Sri Lanka, Bhutan, Maldives…" [36, p181].

**Study design.** Most studies reviewed (n = 23) employed qualitative designs (e.g., ethnography [50,52], grounded theory [19,31,34,35,39], phenomenology [29,47], and case study [17,51]) and analyses [21,23–25,27,30,32,33,43,46,48,53]. There were nine quantitative studies, including retrospective cohort [28,37,42,54] and cross-sectional/survey research [18,38,40,41,49]. There were seven mixed methods studies [16,20,22,26,36,44,45].

**Study settings.** Studies were conducted in several settings that span community, hospital and other settings. Three studies took place in the community hospice environment [31,37,51], ten occurred in the general community/outpatient environment, or through a homecare agency [26,30,32,33,36,38,40,45,50,52]. Three articles reviewed research conducted in the specialized palliative community care environment [19,20,44]. Weber and colleagues worked with health care providers in a specialized palliative care team that followed clients between community and inpatient settings [49], while De Graaff and colleagues explored experiences of health care providers working in community and in the hospital [43]. Fourteen studies took place in the inpatient/hospital setting [16,18,21,23–25,27,28,39,41,42,47,53,54]. There was one study that took place in a long-term care facility [29]. Six studies defined varied study settings, or did not clearly define the location [17,22,34,35,46,48], including Javanparast and colleagues, who performed reviews of intersectoral government documents then interviewed key government stakeholders in their work environment [22].

**Study participants.** The experiences of clinicians, including nurses, physicians and other allied health professionals involved in the provision of palliative and end-of-life care were specifically explored nine studies [19,23,27,36,39,47,49,52,53], while seven others also included the perspectives of patients and families [16,29,31,32,42,43,52]. De Graaff describes this phenomenon as the communication triad, "…triangular communication between the care provider, the patient and a close relative who helped to resolve language problems" [43, p3146]. Zapata and colleagues studied the perspective of both health care providers and patients [42].

There were eight studies that focused solely on the perspective of the patient and their family [18,20,25,28,33,41,45,54]. Three studies focused on the perspective of medical interpreters [21,24,50]. Five studies explored the experiences of administrators and educators involved in end-of-life care [17,34,35,37,48]. For example, Semlali and colleagues aimed to refine training for health professionals in cross-cultural PC and included the perspectives of educators with expertise in ethno-cultural education and training [48]. Another recent study completed a fulsome review of current government policy and legislation that guides PC and enmeshed those findings with the expert opinions of government and policy leaders in their region [22]. The unique perspective of volunteers working in a community hospice was explored in Jovanovic's

study from 2012 [51]. There were five studies that explored the public's ideas around education in palliative and hospice care in linguistically diverse communities [26,30,38,40,44].

**Participant characteristics.** While most studies reported some characteristics of the participants, they rarely reported findings by characteristics, and certainly, none explored the intersection of these characteristics on study results. Overall, age [16,18,20,29,30,32,33,36,38,40,41,44–47,51,52,54], sex [18,20,23–27,30,32,36,39–41,43,44,46,47,49,51,52,54] and occupation [16,19,21–23,27,29,31,32,34–36,39,43,47–50,53] were the most widely reported participant characteristics. Some studies also reported on ethnicity [29,31,36,39,41–44,51] language [18,25,27,28,31,38,40,41,45,49,51], birth country [20,28,30,32,33,38,47,49] or migrant group [26]. Only a few studies reported on the education [32,38,51], and income or socio-economic status [33,54] of participants. One study reported on race [31] and rural/urban status [54]. Broom does not discuss participants in their work [17].

**Terminology and definition of care received.** While there are considerable differences between the role and definition of palliative, hospice, and end-of-life care [1], the terms were often used synonymously and interchangeably within the reviewed studies. For example, Leonard and colleagues uses the term End-of-Life Care in their title but provide definitions that align with the WHO's definition of PC [26]. Palliative Care was the main term used in 16 of the studies examined [17,20,22,24,25,27,28,39,42–44,47–49], but was only clearly defined in ten of those articles [20,22,24,25,39,42,43,47,48,52]; definitions were iterations of the WHO's definition of PC [3]. Kirby and colleagues concede that PC is contextually iterative, not clearly defined, and often open to interpretation [24]. Ten studies used the term end-of-life care [16,19,23,26,29,32,41,46,50,54], only five defined their interpretation end-of-life care [23,26,29,32,54].

Hospice care was explored in six articles [31,34,35,37,40,51]. While two did not clearly define the term [31,34], the remaining had considerable variation in their definitions [35,37,40,51]. For example, Lindley and colleagues defined hospice as a "… model for quality, compassionate care for people facing a life-limiting illness or injury" [37, p2]. Jovanovic defines hospice as care for "…dying patients who had six months or less to live, and who wished to die at home" [51, p165].

Advanced care planning is an important component of end-of-life care, allowing for an individual and their family to clearly define goals-of-care and priorities including limitations of therapy, location of treatment, and expectations around pain and symptom management [1,12]. Four studies focused on advanced care planning and discussions, and the term was clearly and consistently defined in all four studies [18,21,30,38].

Dharni and colleagues studied cancer patients but made specific reference to advanced disease and end-of-life care [45], similar to Glaser and colleagues [33]. Khosla used the term "seriously ill" to describe the health status of the population of interest, but also referred to end-of-life wishes and decision-making [36].

## Synthesis of results

The outcomes studied were categorized into five main themes: (1) impact of language discordance on the quality of PC, (2) culture and care, (3) cultural competence, (4) experience of PC providers, and (5) use of interpreters.

**The impact of language discordance on the quality of palliative care.** Several studies focused on the impact of language discordance on the quality of PC. Non-English speakers often felt neglected and isolated from their care providers because of language discordance, which fostered an environment of mistrust [45]. Individuals who do not speak the same

language as their health care provider have a harder time communicating needs, which limits the quality of assessment by the clinician [28,33]. The length of time in America and degree of language comprehension was found to influence care pathways at end-of-life [41].

Communication barriers contributed to unmanaged pain [25], inadequate symptom management [21], and poor quality of care [51]. Glaser and colleagues found that confusion and misunderstanding after medical appointments led individuals to make decisions they may not have made if they had understood what was being communicated [33]. Individuals with language barriers were also found to be significantly less likely to have advanced directives and resuscitation wishes defined in the context of their care [20], and were more likely to receive aggressive care at the end of life (e.g., cardiopulmonary resuscitation, ventilation, and dialysis) and to die in the intensive care unit [54]. Initial findings indicated that persons with a language barrier had better symptom management, but it was surmised that CALD patients were more likely to under-report their symptoms, giving the appearance of adequate symptom management [28].

**Culture and care.** Cultural context is difficult to grasp and goes beyond the words used to describe a phenomenon. Semlali et al noted that people tend to revert to their mother tongue when experiencing severe illness [48]. In an interview with palliative and hospice care nurses, a nurse talks about the importance of emotional language at the end of life: "you can't fully express yourself, your fears, desires, emotions… unless you can do that in your mother's tongue [23, p273].

Certain cultures have beliefs and practices that may not be fully understood by PC providers. For example, Wardle and Bennett explored the important role that food and feeding played in PC among Punjabi Indians in Australia [29]. Dharni and colleagues described the story of an Islamic patient who could not communicate with their English-speaking medical team the effect of not being able perform their religious ablution had on their health [45].

Family structures are also important. Eckemoff and colleagues identified that there is a common Russian cultural practice of making decisions for sick loved ones, which conflicts with the focus on autonomy in Western societies [32]. A few studies noted how cultural beliefs about death itself played a role in care. For example, some believe that talking about death will cause death, and therefore avoid talking about death altogether [19,52]. Hayes and colleagues quoted a participant discussing advance care planning: "If I'm talking about death, I'm going to actually make it more likely to happen" [21, p4]. Broom and colleagues noted an urgent need for a "… systematic analysis of real-time family negotiations to reveal the complex interplay of culture, language, news telling, clinical judgement and broader negotiations around transitions to the end-of-life" [17, p1045]. Cicolello and Anandarajah highlight the need for a "cultural interpreter" to navigate these complexities [31].

**Cultural competence.** Mislabeling or working under preconceived ideas that a certain group will respond in a certain way can have devastating effects and lead to further barriers to health care delivery [57].

Over half of the studies identified the importance of cultural competence in the context of the delivery of PC [16,20,22,27,29,31,32,34,35,39,43–53]. Green and colleagues found that cultural generalizations made by health care providers contributed to negative patient experiences [20]. Cross-cultural care educators discussed the importance of education for health care providers that highlights self-reflection, positioning individual socio-ethno-cultural backgrounds, and identifying preconceptions as ways to mitigate stereotypes that may be perpetuated by health care providers [48].

**The experience of palliative care providers.** Several studies focused on the impact of linguistic barriers on PC providers. Eckemoff and colleagues found that coordinating and working with interpreters was time consuming and difficult to coordinate [32]. Milberg and

colleagues reported that some nurses described guilt related to the difficulty they experienced in truly empathizing, and being genuine and supportive, to individuals with language barriers [46]. For example: "you have to keep inside things that you would usually say out loud to the patient and their family" [46, p6]. Clinicians have improved job satisfaction when feel they are able to support clients at end-of-life, "assuaging death anxiety through presence, explanations and reassurances" [23, p271]. Researchers also found that nurses working in a well-supported pediatric hospice environment, with manageable workloads, had increased satisfaction in care provided to linguistic minorities, higher use of interpreters, increased use of appropriate educational materials, and better overall client outcomes [37].

**Use of interpreters.** It has been found that providing adequate interpretation services is effective in improving access to hospice for minority groups [34] and leads to better outcomes [31]. Hughes and colleagues reported that most centres had access to remote or in-person translation services, but that these were largely unused [35]; conversely, Green et al reported that health care providers had inadequate access to professional translation services [20]. The cost of translation services, especially for smaller organizations, is a barrier to their use [35]. The study by Green and colleagues reported that medical interpreters were used for discussions of consequence, such as when informed consent was required or a diagnosis was being discussed, and that family members were used for all other visits [19].
However, some have reported the degree of difficulty interpreters have in meaningfully translating medical terminology and jargon. Formal CMIs (or equivalent) have a difficult job that is often criticized in the literature by health care providers and families. Health care providers found medical interpreters a barrier [19]. Some had issues with interpreters who did not translate verbatim [21,26], finding interpreters who did not fully communicate the context of an issue led to an incomplete understanding of care [39]. Some studies report friction between medical interpreters and family members, there was mistrust [43], and different expectations around communication [32]. Kirby and colleagues described the role of the interpreters in end-of-life care as "emotionally taxing, linguistically difficult and highly complex" [24, p498].

Martin and colleagues found that pre-briefings between the interpreter and clinician were helpful, as they clarified expectations and discussed goals prior to meeting with the patient [27]. To that end, guidelines have been developed that provide advice to clinicians to optimize the interpreter's role, and to help clinicians learn about the dilemmas faced by interpreters [50].

There are mixed findings on the family members as interpreters. Some individuals prefer to have their family translate [35], and their presence increases access to (free) interpretation services [17]. However, some clinicians found use of family members as interpreters problematic in that it led to missed opportunities to build rapport with family [19], less than fulsome communication [43], and inconsistent messaging as different family members may interpret differently based on opinion, family structure, and role [32].

**The need for in-language palliative care resources.** Khosla and colleagues found that language-appropriate resources influenced decision-making and that individuals and their families were better equipped to make important decisions for South Asian migrants at the end of life in the United States [36]. Leonard and colleagues found that "it's not just translating a document into a language that's comprehendible. It's translating a whole idea that might now make sense" [26, p8]. Lack of in-language advance care planning resources was a common barrier to members of various CALD communities in Australia [30]. Zapata and colleagues noted that education from clinicians who spoke their language and understood their culture, improved PC utilization by underserved members of the Latinx population [42]. The use of in-language printed resources in pediatric palliative was found to lead to improved experiences and satisfaction [37].

McLean and colleagues studied the impact of sharing language-concordant videos; they found that prior to watching the video, 52% of respondents had no knowledge of advanced directives. After the video, 95% of respondents felt it was important to start to organize end-of-life wishes [38]. Pan and colleagues found that after community education sessions on end-of-life care in hospice, members of the Spanish-, Chinese- and Korean-speaking communities were interested in learning more about hospice care and how it fit in with a palliative approach to care [40].

## Discussion

This scoping review explored 39 studies that investigated PC in linguistic minorities. There was a wealth of studies that originated from North America, which may reflect the current political and social climate of this multicultural region. Several articles were from Australia, where like Canada, there is a push to develop improved approaches to meeting the PC needs of the underserved population, starting with a better understanding of the impact of cultural and linguistic diversity [58]. However, despite a push to improve equitable access to PC in Canada, no Canadian studies were identified that focused on official language minority groups' experiences with PC.

There was an even distribution of research methodologies that explored the phenomenon from the perspective of the service users (patients and their families), frontline clinicians, medical interpreters, and administrators. As such, the body of knowledge contains both information that reflects individual experiences as well as measured outcomes.

In reviewing this incredibly heterogeneous body of literature, there was minimal contextual definition of language or language barriers. While this is an iterative concept, the inconsistency and lack of clear definition makes comparing and generalizing information difficult. Some studies focused on specific languages or ethno-cultural groups, while others discussed the wider concept of language discordance without focusing on a specific population or group.

A palliative approach to care values an individual's right to autonomy and comfort as they progress through a serious illness or condition, regardless of prognosis [59]. When an individual has exhausted options for curative therapies, end-of-life care and hospice services begin, ensuring that predefined goals-of-care for the individual and their family are met [1,3,59]. Despite the differences in palliative, end-of-life and hospice care, there was ambiguity around the PC terminology used, and paucity of studies that focused on a palliative approach to care for folks who are not yet at end-of-life.

Overall, studies found that language discordance has a negative impact on the quality and accessibility of palliative, end-of-life and hospice care, as well as adequate advanced care planning. This was evident from the perspective of patients and their families, and from the perspective of clinicians, interpreters, and health service administrators. This has a significant impact on practice, with numerous suggestions on how administrators and policymakers can adjust policy and procedure to mitigate disparities and provide equitable access to palliative and end-of-life services.

The literature explored the unique and difficult role of interpreters, both formal and informal, and highlighted the importance of barrier-free access to interpretation services. The insights of this diverse body of knowledge can directly inform policy and practice change, highlighting the importance of adequate education, interprofessional collaboration, and cultural competence that helps to meet the needs of underserved community members. Beyond just language, this scoping review identified cultural beliefs, practices and superstitions that may affect the quality of adequate and fulsome language interpretation. The need for language concordant resources, in a variety of media, has been clearly documented to improve access to

palliative and hospice care, as well as adequate advanced care planning and advanced directives, with a goal of decreased deaths in acute care, and increased autonomy and dignity at the end-of-life.

Generalizability was the most common limitation noted within the studies [17–21,24,26,28,29,31,34–36,38,39,42,45–51,54], where study findings may not be relevant outside of the area or population of study. Often, limitations helped to define next steps for many of these studies. Twenty of the studies examined in this scoping review clearly identified further research as a next step to eliciting change [16,17,19,21,25–27,29–31,35,36,39,42,44,45,48,52,54]. Zapata and colleagues identified limitations of generalizability given the limited geographic areas, and lack of specific participant characteristics. They call for further research that involves perspectives from individuals and family and HCPs from a wide swath of geographic and socio-ethno-cultural backgrounds that is more representative of their service area [42].

Many studies had suggestions for practical application, ideas that can guide direct clinical practice change [18,20,22,23,26,34,38,40,41,43]. For example, Pan and colleagues present some clear implications for practice in the hospice environment. Their research clearly demonstrated that Chinese-, Korean- or Hispanic Americans are not averse to having discussions about end-of-life care and hospice services, and encourage PC providers not to avoid these conversations as they have in the past [40].

There is an urgent need to increase accessibility and quality of PC services to underserved communities, including those with language barriers given past and anticipated increases in immigration and diversification in the Canadian population. Research exploring the PC experiences of official language minority communities, as well as allophone communities could help to identify gaps in local service provision and methods to remediate. Taking the lead from circumpolar researchers [50,53], further exploration into the impact of language discordance in those who speak Indigenous languages on the quality of PC could also add depth to the body of literature to support policy and practice change. Lastly, studies that explore a palliative approach to care before end-of-life could also help identify ways to increase knowledge and utilization of this important service.

## Limitations

This scoping review provides a broad understanding of the literature in the field, rather than an in-depth analysis of a particular issue, it did not assess the quality and risk of bias in the articles reviewed; these are general limitations of scoping reviews [60]. Though the selection and extraction were not completed by two independent reviewers, it is believed that the process utilized was both rigorous and sufficient to ensure the validity of the work.

## Conclusion

Employing an outcomes-oriented focus, this scoping review focused on studies that explore the outcomes of interventions that improve the PC experiences for individuals with language barriers. This literature review has met its objectives to examine PC in the context of linguistic minorities, explore the range and scope of the existing research related to the phenomenon, and identify and synthesize important trends and variables in the research as they emerged from the literature. The narrative described research findings that highlight disparities in PC experiences in language discordant care delivery, analyzed important trends in the literature, identified areas for further research, and discussed impact on current practice and policy.

## Supporting information

**S1 Checklist.** PRISMA-ScR Checklist Feb 26.
(DOCX)

## Author contributions

**Conceptualization:** Shawn P. Dookie, Lynn Martin.

**Data curation:** Shawn P. Dookie.

**Formal analysis:** Shawn P. Dookie, Lynn Martin.

**Funding acquisition:** Lynn Martin.

**Resources:** Lynn Martin.

**Validation:** Lynn Martin.

**Writing – original draft:** Shawn P. Dookie.

**Writing – review & editing:** Shawn P. Dookie, Lynn Martin.

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
