## [Decision Letter · Decision Letter 0]

12 Jan 2025

PONE-D-24-51369The Effect of Language Discordance on the Experience of Palliative Care: A scoping reviewPLOS ONE

Dear Dr. Dookie,

Thank you for submitting your manuscript to PLOS ONE. After careful consideration, we feel that it has merit but does not fully meet PLOS ONE’s publication criteria as it currently stands. Therefore, we invite you to submit a revised version of the manuscript that addresses the points raised during the review process.

Specifically, the rationale for conducting a scoping review should be explained in the Methods section to clarify its suitability for exploring the breadth of literature and identifying gaps in research on language discordance in palliative care. Additionally, the description of how article selection decisions were made, currently located in the Limitations section, is better moved to the Methods section. This should include details on the criteria used, the decision-making process, and how disagreements between authors were resolved. Finally, the Methods section should provide more details on the data charting and extraction process, e.g., specifying whether the process involved more than one researcher, whether the data extraction spreadsheet was pilot-tested, and a brief overview of the piloting process.

We look forward to receiving your revised manuscript.

Kind regards,

Weifeng Han, PhD

Academic Editor

PLOS ONE

“The production of this scoping review has been made possible through a financial contribution from Health Canada. The views expressed herein are the views of the authors, and do not necessarily represent the views of Health Canada.”

Reviewers' comments:

Reviewer's Responses to Questions

**Comments to the Author**

1. Is the manuscript technically sound, and do the data support the conclusions?

Reviewer #1: Yes

Reviewer #2: Yes

2. Has the statistical analysis been performed appropriately and rigorously? 

Reviewer #1: N/A

Reviewer #2: N/A

3. Have the authors made all data underlying the findings in their manuscript fully available?

Reviewer #1: Yes

Reviewer #2: Yes

4. Is the manuscript presented in an intelligible fashion and written in standard English?

Reviewer #1: Yes

Reviewer #2: Yes

5. Review Comments to the Author

Reviewer #1: The issue of access to palliative care for people from diverse languages and backgrounds is an important one. This paper focuses on where there is discordance between the professional caregivers and patients and the problems that might arise for good quality palliative care (or indeed receiving any palliative care). It is a very useful review article that is clearly written and comprehensive. I look forward to its publication so I can use it in my own research.

Reviewer #2: Review of PONE-D-24-51369 Effects of Language Discordance on the Experience of Palliative Care: A Scoping Review

This scoping review examines articles relating to language discordance in palliative care, published between Jan 2010 and Feb 2024, in English and French. It focuses on the experiences of palliative care recipients whose language does not match that of their care providers. Based on the literature, language discordance appears to have negative effects on palliative care, end of life care, hospice care, and advance care planning. This scoping review can prove a valuable addition to the literature. There are a few details that would strengthen the paper.

1. What was the rationale for conducting a scoping review rather than another type of knowledge synthesis?

2. The information on which authors made decisions on article selection was presented in the Limitations section. A brief description should be placed in the Methods section. How were article selection decisions made? How were differences in decisions between the authors resolved?

3. Please add details on the data charting and extraction process. Was this done by one researcher or more? Was the data extraction spreadsheet pilot tested? Please describe that process.

6. PLOS authors have the option to publish the peer review history of their article (what does this mean? ). If published, this will include your full peer review and any attached files.

**Do you want your identity to be public for this peer review?** For information about this choice, including consent withdrawal, please see our Privacy Policy .

Reviewer #1: **Yes: ** Adjunct Professor Rosemary Jill Leonard

Reviewer #2: No

---

## [Author Response · Author response to Decision Letter 1]

5 Feb 2025

We were pleased with the first reviewer’s very positive feedback on our manuscript, as well as with the thoughtful comments of the second reviewer. Below, you will see the reviewers’ comments in bold, followed by our responses and location of changes in the revised manuscript. We believe that these revisions have greatly strengthened the manuscript.

1. What was the rationale for conducting a scoping review rather than another type of knowledge synthesis?

We have revised the text to present the rationale and justification for conducting a scoping review (versus a systematic review); see Lines 107-111 at the very beginning of the “Methods” section.

Synthesis of the literature was conducted using a scoping approach, as it aligned well with our research goal of exploring and mapping out the current literature. With the goal of identifying and synthesizing the broad, heterogeneous range of study designs and findings, our scoping review will be used to inform future research priorities in language discordance in palliative care [14, 15]. To provide a more structured methodological approach, the Preferred Reporting Items for Systematic reviews and Meta-Analyses extension for Scoping Reviews (PRISMA-ScR) Checklist was used to guide reporting of the scoping review procedures and findings [15].

2. The information on which authors made decisions on article selection was presented in the Limitations section. A brief description should be placed in the Methods section. How were article selection decisions made? How were differences in decisions between the authors resolved?

We have added details to the manuscript, as suggested by the reviewer; see Lines 128-134 under ‘Eligibility Criteria’ in the Methods section.

The first author led the search, selection process, and data extraction with the support of a Librarian and the second author. Using the mutually developed search terms, the first author performed initial searches and title reviews, documenting reasons for exclusion. The second author reviewed all decisions; any disagreements were discussed and decisions (to include or exclude) were approved by both. The same process was followed through the abstract reviews. We did not experience any disagreements in decision-making and tended to air on the side of inclusion if we questioned the relevance of an article.

3. Please add details on the data charting and extraction process. Was this done by one researcher or more? Was the data extraction spreadsheet pilot tested? Please describe that process.

We agree with this suggestion and have added details related to the initial data items for extraction as well as author roles and the data charting/extraction process; see Lines 151-161 under “Data Charting and Data Items” in the Methods section.

Data charting and extraction was completed using a Microsoft Excel Spreadsheet. This scoping review is part of a much larger project exploring palliative the care experiences of underserved populations, and as such, we initially worked with a pre-determined set or common set of data items for extraction: country; target population; study design, setting, participants, and characteristics; terminology and definition of care received; outcomes studied; and author-defined study limitations and next steps. The first author completed the extraction for the first ten articles (in alphabetical order by first author), and these were fully reviewed by the second author and no changes were made to the extraction. The second author completed extraction of the remaining articles. The first author identified a set of initial codes for summarizing key elements in the extracted information, which were reviewed by the second author. Inconsistences and discrepancies identified were discussed and resolved to the satisfaction of both authors. The data extraction summary included below represents the final collaborative findings of both authors.

---

## [Decision Letter · Decision Letter 1]

23 Feb 2025

PONE-D-24-51369R1The Effect of Language Discordance on the Experience of Palliative Care: A scoping reviewPLOS ONE

Dear Dr. Dookie,

Thank you for submitting your revised manuscript to PLOS ONE and for your helpful responses to the previous reviewer comments. I have taken over as editor of your paper and have just one further suggestion for you to consider. On first reading your abstract, I was unsure why you were focusing on Canada in the conclusion. I then started to wonder if the scoping review only considered Canadian literature, despite the title of the paper. On reading the full manuscript, the answer became clear - but most readers will not get beyond the abstract. Your covering letter explains that the study was originally designed to explore Canadian official language minority communities but was broadened for a more robust dataset. Perhaps some wording along these lines could be included earlier in the abstract so that it is clear that your results have a relevance wider than just Canada? Therefore, we invite you to consider submitting  a revised version of the manuscript that addresses this point.

We look forward to receiving your revised manuscript.

Kind regards,

Antony Bayer

Academic Editor

PLOS ONE

Journal Requirements:

Reviewers' comments:

Reviewer's Responses to Questions

**Comments to the Author**

1. If the authors have adequately addressed your comments raised in a previous round of review and you feel that this manuscript is now acceptable for publication, you may indicate that here to bypass the “Comments to the Author” section, enter your conflict of interest statement in the “Confidential to Editor” section, and submit your "Accept" recommendation.

Reviewer #2: All comments have been addressed

2. Is the manuscript technically sound, and do the data support the conclusions?

Reviewer #2: Yes

3. Has the statistical analysis been performed appropriately and rigorously? 

Reviewer #2: N/A

4. Have the authors made all data underlying the findings in their manuscript fully available?

Reviewer #2: Yes

5. Is the manuscript presented in an intelligible fashion and written in standard English?

Reviewer #2: Yes

6. Review Comments to the Author

Reviewer #2: (No Response)

7. PLOS authors have the option to publish the peer review history of their article (what does this mean? ). If published, this will include your full peer review and any attached files.

**Do you want your identity to be public for this peer review?** For information about this choice, including consent withdrawal, please see our Privacy Policy .

Reviewer #2: No

---

## [Author Response · Author response to Decision Letter 2]

26 Feb 2025

We agree with your feedback, and took your advice to amend our abstract, to better reflect that our scoping review was quite broad and widely incorporated international data. You will now find this sentence, on line 36, in the “Design” section of our abstract: “This scoping review was originally designed to explore Canadian official language minority communities but was broadened to an international search for a more robust dataset.”

Again, we thank you for the opportunity to submit minor revisions to our manuscript. We believe that the revised manuscript addresses the issues you have raised. We look forward to hearing from you on our revised manuscript and to working with you toward its publication.

---

## [Editor Report · Decision Letter 2]

2 Mar 2025

The Effect of Language Discordance on the Experience of Palliative Care: A scoping review

PONE-D-24-51369R2

Dear Dr. Dookie,

We’re pleased to inform you that your manuscript has been judged scientifically suitable for publication and will be formally accepted for publication once it meets all outstanding technical requirements.

Kind regards,

Antony Bayer

Academic Editor

PLOS ONE
---

## [Editor Report · Acceptance letter]

PONE-D-24-51369R2

PLOS ONE

Dear Dr. Dookie,

I'm pleased to inform you that your manuscript has been deemed suitable for publication in PLOS ONE. Congratulations! Your manuscript is now being handed over to our production team.

Kind regards,

on behalf of

Professor Antony Bayer

Academic Editor

PLOS ONE